# CBP/β-Catenin/FOXM1 Is a Novel Therapeutic Target in Triple Negative Breast Cancer

**DOI:** 10.3390/cancers10120525

**Published:** 2018-12-19

**Authors:** Alexander Ring, Cu Nguyen, Goar Smbatyan, Debu Tripathy, Min Yu, Michael Press, Michael Kahn, Julie E. Lang

**Affiliations:** 1Department of Oncology and Hematology, UniversitätsSpital Zürich, Rämistrasse 100, 0832 Zürich 1, The Netherlands; alexander.ring@usz.ch; 2Department of Molecular Medicine, Beckman Research Institute, City of Hope, Duarte, CA 91010, USA; CNGUYEN3@msn.com (C.N.); smbatyan@usc.edu (G.S.); 3Department of Breast Medical Oncology, UT-MD Anderson Cancer Center, Houston, TX 77030, USA; DTripathy@mdanderson.org; 4Department of Stem Cell Biology and Regenerative Medicine, University of Southern California, Los Angeles, CA 90033, USA; minyu@med.usc.edu; 5University of Southern California Norris Comprehensive Cancer Center, Los Angeles, CA 90033, USA; Michael.Press@med.usc.edu; 6Department of Surgery, University of Southern California, Los Angeles, CA 90033, USA

**Keywords:** triple negative breast cancer (TNBC), cancer stem cells (CSC), CREB-binding protein (CBP), forkhead box protein M1 (FOXM1), ICG-001

## Abstract

*Background:* Triple negative breast cancers (TNBCs) are an aggressive BC subtype, characterized by high rates of drug resistance and a high proportion of cancer stem cells (CSC). CSCs are thought to be responsible for tumor initiation and drug resistance. cAMP-response element-binding (CREB) binding protein (CREBBP or CBP) has been implicated in CSC biology and may provide a novel therapeutic target in TNBC. *Methods:* RNA Seq pre- and post treatment with the CBP-binding small molecule ICG-001 was used to characterize CBP-driven gene expression in TNBC cells. In vitro and in vivo TNBC models were used to determine the therapeutic effect of CBP inhibition via ICG-001. Tissue microarrays (TMAs) were used to investigate the potential of CBP and associated proteins as biomarkers in TNBC. *Results:* The CBP/ß-catenin/FOXM1 transcriptional complex drives gene expression in TNBC and is associated with increased CSC numbers, drug resistance and poor survival outcome. Targeting of CBP/β-catenin/FOXM1 with ICG-001 eliminated CSCs and sensitized TNBC tumors to chemotherapy. Immunohistochemistry of TMAs demonstrated a significant correlation between FOXM1 expression and TNBC subtype. *Conclusion:* CBP/β-catenin/FOXM1 transcriptional activity plays an important role in TNBC drug resistance and CSC phenotype. CBP/β-catenin/FOXM1 provides a molecular target for precision therapy in triple negative breast cancer and could form a rationale for potential clinical trials.

## 1. Introduction

Breast cancer (BC) is a heterogeneous disease [1], yet clinical management is based on only three biomarkers: estrogen receptor (ER), progesterone receptor (PR) and the human epidermal growth factor receptor (HER2) [2]. About 15% of all BCs are negative for all three markers (triple negative–TN). TNBCs are characterized by aggressive growth, high rates of drug resistance and poor survival rates [3]. On the molecular level, the majority of TNBCs have basal-like gene expression patterns [4] and over 90% of basal-like BCs are TN [5]. TNBCs contain higher numbers of cancer stem cells (CSCs) [6] and also exhibit stem-like gene expression signatures [7]. CSCs are thought to be responsible for tumor initiation, drug resistance and metastasis [8]. On the molecular level, nuclear β-catenin-driven transcription has been shown to play an important role in CSC biology [9,10]. CREB-binding protein (CBP) and/or E1A-associated protein p300 (p300) are important co-activators in β-catenin-driven transcription [11] and play critical roles in various cellular functions such as proliferation, cell cycle regulation and apoptosis [12]. Despite their high degree of homology, both proteins have distinct functions [13], in particular in stem cell biology, either maintaining an undifferentiated state or inducing differentiation, respectively [14]. CBP has been shown to play an important role in CSC biology and provides a critical target in cancer [15]. As a co-activator, CBP regulates the transcriptional activity of FOXM1 [16]. 

FOXM1 is one of the most commonly up-regulated transcripts in various cancers, including breast cancer [17]. It is critically important for cell cycle progression [18] and plays a key role in therapy resistance [19]. FOXM1 has been implicated in breast cancer CSC phenotype [20] and TNBC/basal-like BC biology [1,21]. ICG-001 is a specific CBP-binding small molecule [22] and has been shown in pre-clinical models to target CSC populations and sensitize cancer cells to chemotherapy [15,23]. In the current study, we identified the CBP/β-catenin/FOXM1 transcriptional complex as an important driver in TNBC biology and a potential novel therapeutic target.

## 2. Results

### 2.1. CBP as a Potential Target in TNBC

CBP amplification was identified in 17.7% of BCs (mean 9.92 ± 6.44%) (Figure 1A). Increased expression of CBP was observed in BC compared to normal breast and in TNBC compared to other BC subtypes in the TCGA data set (Figure 1B). CBP overexpression was also observed in the METABRIC data set (fold change 1.16, *p* = 0.004) [24]. Alterations in p300 were also present in BC, albeit at significantly lower levels (e.g., amplification 0.32 ± 0.11%) (Appendix A). Protein levels of CBP were high in TNBC cell lines (MDA-MB-231 and MDA-MB-468) compared to the non-tumorigenic breast epithelial cell line MCF10a (Figure 1C). Previous studies demonstrated that survivin (BIRC5) is a direct target of CBP/β-catenin transcription [13]. Survivin was highly expressed in MDA-MB-231 and MDA-MB-468 cells, compared to MCF10a (Figure 1C). Co-Immunoprecipitation (CoIP) demonstrated that CBP binds to β-catenin in three TNBC cell lines (MDA-MB-231, MDA-MB-468 and SUM149) under DMSO control conditions, which can be disrupted with 20μM ICG-001 (Figure 1D). Treatment with ICG-001 led to the down-regulation of survivin reporter activity (Figure 1E) and protein levels (Figure 1F). ICG-001 specifically inhibits the viability of CBP-dependent MDA-MB-231 cells, but not non-transformed MCF10a cells (Figure 1G).

### 2.2. FOXM1 is a Downstream Effector of CBP-Signaling in TNBC

CBP/β-catenin form transcriptionally active complexes via interaction with DNA-binding TFs [25,26]. Differential gene expression analysis of whole transcriptome RNA Seq data of MDA-MB-231 treated for 48 h with either 10 μM ICG-001 or DMSO vehicle control revealed that 1339 genes are differentially expressed between treatment and control conditions (DMSO vs. ICG-001 729 genes up-regulated, 610 genes down-regulated, FDR ≤ 0.05, ≤ |2-fold| change) (Figure 2A). Analysis of this differentially expressed gene-signature using Ingenuity Pathway Analysis (IPA) revealed FOXM1 as a potential upstream-regulator of the gene expression changes observed (Figure 2B). The TCGA BC RNA Seq dataset confirmed that TNBCs are characterized by high expression of FOXM1 target genes compared to other molecular subtypes (Appendix A). Comparison of CBP and FOXM1 RNA expression in the TCGA BC (all subtypes) and TNBC datasets via Oncomine showed that 39.5% (30/76) and 33.3% (15/45) of samples with higher FOXM1 expression had higher CBP expression, respectively (Figure 2C). The TCGA data set further confirmed that FOXM1 RNA expression is higher in BC tissue compared to normal breast and higher in TNBC compared to other subtypes (Figure 2D). Higher FOXM1 expression was also observed in the METABRIC data set (fold change 2.21, *p* = 4.54 × 10^−149^) [24].

### 2.3. CBP, FOXM1 and β-Catenin form a Transcriptionally Active Complex That Can be Targeted with ICG-001

CoIP demonstrated that FOXM1 was associated with CBP (Figure 3A) and β-catenin (Figure 3C) in MDA-MB-231 cells. Treatment with ICG-001 for 4 h and 24 h led to a reduction of FOXM1 in complex with CBP (Figure 3A,B, respectively), but not β-catenin (Figure 3C), supporting the previously reported specific binding of ICG-001 to CBP [22]. Protein levels of FOXM1 were not affected after 4 h (Figure 3D), but showed a strong reduction after 24 h (Figure 3E). The Kahn lab has previously demonstrated that ICG-001 does not affect the levels of CBP or β-catenin [22]. ICG-001 treatment also reduced FOXM1 protein levels in MDA-MB-468 cells after 24 h (Figure 3F).

RNA Seq analysis showed that the expression of the majority of FOXM1 target genes found in the TCGA dataset was reduced in MDA-MB-231 cells treated with ICG-001 (Figure 4A) and down-regulation of selected targets was confirmed via RT-qPCR (Figure 4B). Concordantly, activity of a FOXM1-driven firefly luciferase reporter was significantly reduced after 24 h treatment with ICG-001 compared to DMSO control in three different TNBC cell lines (MDA-MB-231, MDA-MB-468, Hs578T) (Figure 4C). FOXM1 target gene expression was also reduced in three additional TNBC cell lines (MDA-MB-468, MDA-MB-453, Hs578T) (Appendix A). Transient transfection with siRNAs directed against CBP, FOXM1 and β-catenin demonstrated that simultaneous Knockdown (KD) of FOXM1 and β-catenin had the strongest effect on FOXM1 target gene expression (Figure 4D and Appendix A).

### 2.4. ICG-001 Ameliorates Paclitaxel-Induced Increases in FOXM1 Expression and CSC Phenotype as Well as Tumor Initiation Capacity In Vitro

Treatment of MDA-MB-468 TNBC cells with 10 nM paclitaxel led to an initial reduction of cell numbers after 48 h, with recurrence of cell colonies after five to seven days despite the continued presence of paclitaxel (Figure 5A). These paclitaxel-resistant MDA-MB-468 cells showed increased levels of FOXM1 protein expression compared with treatment naïve cells (Figure 5B), significantly higher numbers of CSC like CD24_low_CD44^high^ (Figure 5C) and side-population cells (Figure 5D) (*p* < 0.0001) and overexpression of stem cell and drug resistance markers (Figure 5E). Treatment of MDA-MB-468 with 10nM paclitaxel plus 10 μM ICG-001 abolished the occurrence of drug resistant cell colonies compared to paclitaxel treatment alone (DMSO 31 ± 4 colonies vs. ICG-001 0 colonies, *p* < 0.05) (Figure 5F). Retreatment of paclitaxel resistant CSC-like MDA-MB-468 cells (post 5 d 10 nM paclitaxel treatment) with a combination therapy of 10 nM paclitaxel plus 10 μM ICG-001 led to a significant reduction of paclitaxel resistant cells, while a re-challenge with paclitaxel alone had no effect (DMSO 29 ± 4 colonies vs. ICG-001 3 ± 1, 8.9-fold reduction, *p* < 0.05) (Figure 5G). Treatment with ICG-001 also reduced the expression of stem cell and drug resistance marker genes (Figure 5E). The number of cells with side population phenotype was strongly reduced in MDA-MB-231, MDA-MB-468, as well as in TNBC cells isolated from a patient derived xenograft (PDX) (Figure 5H). Tumorsphere formation was used as a surrogate for tumor initiation capacity in vitro [27]. MDA-MB-231 cells continuously treated with ICG-001 for nine days showed a statistically significantly reduced number of spheres compared to vehicle control treated cells (*p* < 0.001) (Figure 6A). Transient siRNA-mediated KD of FOXM1, β-catenin and CBP demonstrated a reduction in sphere formation compared to non-specific control (scramble) siRNA transfected cells (*p* < 0.05) (Figure 6B). ICG-001 also significantly reduced the number of spheres in CAL51, MDA-MB436, Hs578T and SUM149 cell lines (Figure 6C and Appendix A).

### 2.5. Targeting CBP Enhances the Response to Paclitaxel In Vivo Dependent on FOXM1 Expression

Mice bearing MDA-MB-468 cell line xenografts and treated with paclitaxel plus ICG-001 showed a statistically significantly reduced tumor burden compared to either paclitaxel or ICG-001 alone, as well as compared to PBS vehicle control treated animals (Figure 7A and Appendix A). To demonstrate the presence of tumor initiating cells after treatment, tumors were removed, dissociated and implanted into healthy female NSG mice, without further administration of any treatment. The tumors previously treated with paclitaxel plus ICG-001 developed statistically significantly smaller secondary tumors (Figure 7B and Appendix A).

PDX mouse models from two TNBC patients were established: PDX1 was from a tumor with low FOXM1 protein expression and a small fraction of side population cells (0.2%), while PDX2 was from a tumor with relatively higher expression of FOXM1 and higher proportion of side population cells (5%) (Figure 7C). Clinical data available for both patients showed that despite similar tumor grade and stage as well as treatment received, the patient whose tumor expressed high levels of FOXM1 had a worse survival outcome (Appendix A). Accordingly, tumors in mice bearing PDX2 showed more aggressive growth (Appendix A) and worse response to therapy. PDX1 tumors responded similarly to paclitaxel alone or combination of paclitaxel plus ICG-001 (Figure 7D and Appendix A) upon initial treatment and inhibition of outgrowth upon secondary implantation without further treatment (Figure 7E and Appendix A). In mice bearing PDX2 on the other hand, only paclitaxel plus ICG-001 resulted in a statistically significant reduction in primary tumor growth (Figure 7F and Appendix A) and reduced secondary engraftment and outgrowth (Figure 7G and Appendix A). Quantification of FOXM1 and ABCG2 in PDX2 bearing mice post treatment showed reduced protein expression after treatment with paclitaxel plus ICG-001 (Appendix A). These data suggest that targeting CBP/FOXM1 via the small molecule inhibitor ICG-001 enhanced the initial response and duration of remission after paclitaxel chemotherapy in vivo.

### 2.6. Tissue Micro Arrays (TMA) Stained for FOXM1 Demonstrated Correlation with TNBC Subtype and Tumor Grade

FOXM1 protein expression was significantly correlated with TNBC subtype (*p* < 0.001), ER/PR+ tumors (*p* = 0.045) and tumor grade (*p* = 0.013) by chi squared analysis (Appendix A). A significant correlation was found between strong (2) CBP staining and HER2 positive status (*p* = 0.017) as well as positive lymph node metastasis (N1) (*p* = 0.034) (Appendix A). The BC subtype distribution of the TMAs is shown in Appendix A. Overall scoring results for FOXM1 and CBP are shown in Appendix A. Appendix A shows representative cases for FOXM1 and CBP staining on the TMAs. Multivariate logistic regression analysis showed a statistically significant association between high FOXM1 protein levels and TNBC subtype (*p* = 0.034), tumor grade II (*p* = 0.023) and III (*p* = 0.01) and radiation therapy (*p* = 0.04) (Appendix A). Calculation of hazard ratios using a Cox regression model showed unfavorable HRs for TNBC subtype (HR 1.63), tumor stage 2 (HR 1.43) and tumor grade III (HR 1.63), while a favorable HR (0.035) was found for age ≤ 50 years (Appendix A). FOXM1 and high tumor grade showed an AUC of 0.75 in predicting TNBC (Appendix A). Kaplan-Meier analysis of the TMA data for association between FOXM1 levels and 5-year overall survival (OS) demonstrated no association in all BC cases (*p* = 0.14) (Appendix A) or TNBC only (*p* = 0.83) (Appendix A). CBP protein expression and OS similarly showed no statistically significant correlation between outcome and expression (all cases *p* = 0.096, TNBC *p* = 0.30) (Appendix A). Using TCGA RPPA BC data suggested a trend between FOXM1 expression and worse 5-year OS (*p* = 0.059) for all cases (Appendix A), but not for TNBC (*n* = 82) (*p* = 0.21) (Appendix A). No RPPA data was available for CBP.

## 3. Discussion

CBP is an important co-activator in β-catenin driven transcription, which has been implicated in CSC [9,10] and TNBC biology [28]. The disruption of CBP/β-catenin with ICG-001 effectively targets various cancers by eliminating CSCs [23,29,30]. Since neither CBP nor β-catenin bind directly to DNA they form a transcriptionally active complex with DNA binding TF’s [16,25]. The current study identified CBP/β-catenin/FOXM1 as an important driver of CSC phenotype in TNBC.

FOXM1 has been shown to be an important factor in breast cancer CSC phenotype [20] and TNBC/basal BC biology [1,21]. Direct targeting of TF’s such as FOXM1 via small molecule inhibitors poses significant challenges (e.g., lack of substrate binding pockets) [31]. Gene KD via siRNA directed against FOXM1 has been explored in vitro [32] and in vivo [33], but the clinical application of this approach is limited by off-target effects or severe inflammatory reactions [34]. The proteasome inhibitor thiostrepton targets FOXM1 indirectly [35], but also inhibits mitochondrial translation non-specifically [36]. Indirect targeting of protein-protein interactions between TF’s and co-activators via specific small molecule inhibitors presents a potential alternative approach [37].

CBP has been previously shown to function as a transcriptional co-activator in FOXM1-driven transcription [16]. The current study demonstrated that ICG-001 disrupted CBP/β-catenin/FOXM1, resulting in reduced expression of FOXM1 and FOXM1 target genes. However, CoIP of CBP/FOXM1 and the fact that ICG-001 decreased this interaction does not prove that this interaction is direct. ICG-001 was discovered by screening a secondary structure-template small molecule library and was shown to specifically bind to the 1–111 amino acid region of CBP, thereby competing for β-catenin binding [22]. FOXM1 binds more C-terminally to CBP than ICG-001; therefore ICG-001 probably does not affect FOXM1/CBP binding directly. The data presented here suggested that ICG-001 disrupts CBP/β-catenin, as has been previously demonstrated [22], thereby preventing FOXM1/β-catenin bridging to the basal transcriptional apparatus and disrupting FOXM1 driven transcription. This is also consistent with the inability of ICG-001 to disrupt the FOXM1/β-catenin interaction. Halasai et al. proposed that FOXM1 levels are regulated via an auto-regulatory loop in which FOXM1 binds to its own promoter in a positive feedback loop [38]. We identified two FOXM1 consensus binding sites (A(T/C)AAA(T/C)AA) in the proximal (1 kb) promoter region of FOXM1 (unpublished data). Chromatin immunoprecipitation could be used to investigate CBP, β-catenin and FOXM1 binding to these sites and the effect of ICG-001 treatment. siRNA knock down of FOXM1 plus β-catenin had the strongest affect on FOXM1-driven gene expression. A possible explanation could be that gene or protein dosage of FOXM1 and β-catenin is more critical to their transcriptional activity than CBP, while the interaction of CBP with FOXM1 and β-catenin is critical for a functional transcriptional complex. Hence, as siRNA knockdown cannot fully eliminate CBP from the cells, enough of the protein might be left to drive CBP/FOMX1/β-catenin dependent gene expression. On the other hand, treatment with ICG-001 potently disrupts the binding of CBP to the transcriptional complex, thereby inhibiting gene expression.

The current study demonstrated that ICG-001 eradicates CSC populations in TNBC, as had been previously reported for other cancers [15,23,39]. It is interesting that in TNBC cells combined KD of CBP and β-catenin had the strongest effect on mammosphere formation as a surrogate for CSC phenotype, and best recapitulated the effect of ICG-001. These results further supported the notion that ICG-001 blocked CBP/catenin binding, and thereby indirectly disrupted FOXM1 transcriptional activity, but also potentially effected the interaction of other TFs with CBP/β-catenin that may be important for the CSC phenotype. 

The current study showed that the CSC phenotype is increased in paclitaxel resistant TNBC cells, which was associated with increased FOXM1 levels. Treatment with ICG-001 in combination with paclitaxel prevented the occurrence of drug resistant CSC-like cell population and resensitized pretreated resistant cells to paclitaxel. The paclitaxel resistant phenotype was one of rapid resistance and more likely associated with adaptive responses as opposed to clonal selection. Future clonogenic approaches could further elucidate a CBP/β-catenin/FOXM1-dependent drug resistant CSC phenotype in TNBC. Since ICG-001 does not cause cytotoxicity, a potential mechanism by which CSC are eliminated could be through forced differentiation. Differential co-activator usage of CBP and p300 have been shown to affect stem cell fate [14], providing a potential mechanism that could be further explored in TNBC CSCs.

The in vivo response to treatment in the primary tumors fit the proposed dependence on FOXM1 levels (i.e., PDX1 with low FOXM1 responded similarly to paclitaxel alone or combination treatment with ICG-001, while the MDA-MB-468 xenograft and PDX2 with higher levels of FOXM1 showed the greatest reduction in tumor growth by combination treatment with paclitaxel plus ICG-001). Yet the behavior observed in the secondary outgrowth, which was used as a surrogate for disease recurrence [40], was more varied. Likely a more complex environment in vivo compared to in vitro culture affects the response to treatment. Treatment with ICG-001 resulted either in large tumors or prevented secondary outgrowth. It could be speculated that ICG-001 can either drive CSC towards a transiently amplifying phenotype that still has tumor initiating capacity but is more sensitive to paclitaxel treatment, or differentiate CSC to a level where these cells loose tumor initiating capacity. Especially in vivo this outcome might be determined by drug concentration within tissue, or modified by a complex tumor microenvironment.

Although in vitro treatment with ICG-001 effectively targeted drug resistant CSC-like cells, residual secondary outgrowth was observed to a varying degree in vivo. One possible explanation could be the development of resistance to ICG-001 under prolonged treatment, although resistance has not been observed in several clinical trials (NCT01764477, NCT01606579). Other factors such as drug bioavailability (e.g., caused by poor vascularization) might contribute to the observed differences. Re-treatment of dissociated tumors in vitro could further elucidate the underlying mechanisms. 

FOXM1 has previously been associated with various clinical characteristics in BC, such as tumor stage and nodal status [41], ER+ status [42] as well as TN subtype [41]. In this study, TMAs confirmed the association of high FOXM1 protein levels with TNBC subtype and high tumor grade. Using FOXM1 and tumor grade as biomarkers can predict TNBC subtype with 75% accuracy. The correlation of higher FOXM1 levels with increased CSC populations and resistance to chemotherapy in combination with the ameliorating effect of targeting FOXM1 via the CBP-binding small molecule ICG-001 warrants further exploration of FOXM1 as a predictive biomarker.

## 4. Materials and Methods

### 4.1. Cell Culture

MDA-MB-231, MDA-MB-468, MDA-MB-436 and Hs578T were obtained from the American Type Culture Collection (ATCC) (Manassas, VA, USA). SUM149 was obtained from Asterand Bioscience (Detroit, MI, USA). CAL51 was obtained from The Leibniz Institute DSMZ-German Collection of Microorganisms and Cell Cultures GmbH (Braunschweig, Germany). All cell lines were authenticated and tested for mycoplasma contamination. Appendix A lists the culture conditions for all cell lines. As a surrogate for tumor initiation capacity [27], TNBC cells were cultured at a density of 20,000 cells per well in Mammocult medium (STEMCELL Technologies, Vancouver, Canada) using ultra-low adherence 6-well culture plates (Corning Inc., Corning, NY, USA) for 7 to 9 days. Tumorspheres were counted based on size (small < 50 μm, large > 50 μm). Paclitaxel-resistant MDA-MB-468 triple negative BC cells were selected via continuous treatment with 10nM paclitaxel for 5 to 7 days, at which time drug resistant cell colonies emerged. Cell viability in vitro was assessed using the Bright Glo Luciferase assay (Promega, Madison, WI, USA).

### 4.2. Transient Transfection and siRNA-Mediated Gene Knockdown (KD)

Lipofectamine 2000 Reagent (Thermo Fisher Scientific, Canoga Park, CA, USA) was used according to the manufacturer’s protocol for transfection. The survivin and FOXM1 reporter plasmids (pGL3b-6270-survivin-luciferase or pGL3b-FOXM1-luciferase, respectively) and pRL3 Renilla luciferase control vector (Promega) were used. Transfected cells were treated after 24 h transfection for 24 h with 10 μM ICG-001 or DMSO vehicle control. Luciferase reporter activity was quantified using the Dual-Glo Luciferase Assay System (Promega) and a PerkinElmer EnVision Multilabel plate reader (PerkinElmer, Waltham, MA, USA). 

All siRNA transient gene KDs for FOXM1, CBP, β-catenin and negative control were performed using Silencer Select siRNAs (Thermo Fisher Scientific) and Lipofectamine RNiMAX (Thermo Fisher Scientific) according to the manufacturer’s protocol. Gene KD was validated via RT-qPCR 48 h after transfection.

### 4.3. Xenografts of TNBC

MDA-MB-468 cells were prepared in a 1:1 mix of cell culture medium (DMEM/ F12) and Matrigel (BD Biosciences, San Jose, CA, USA). 1 × 10^6^ cells were injected into the hind flanks of female NOD-scid IL2rγ^null^ (NSG) mice. All experimental procedures involving mice were approved by the Institutional Animal Care and Use Committee at the University of Southern California (USC) (IACUC protocol 11204). For PDX models, surgical specimens from patients with TNBC were obtained with the permission of the Institutional Review Board (HS-16-00379) of USC and were collected immediately after surgery. Tumor pieces (3 mm^3^) were placed into Matrigel and implanted into the upper mammary fat pads of female NSG mice. Tumor growth was monitored weekly using a digital caliper (VWR, Radnor, PA, USA). Tumor volume was calculated using the formula Volume = (Width × 2 × Length)/2 [43].

### 4.4. Drug Application and Treatment Conditions

For in vitro application, paclitaxel (SupremeMed, Van Nuys, CA, USA) and ICG-001 were prepared in dimethyl sulfoxide (DMSO). In vivo treatment of xenograft bearing mice was initiated after palpable tumors developed. Paclitaxel was prepared in sterile PBS to yield a final concentration of 10 μg/ KG body weight and applied per mouse weekly via intraperitoneal injection.

2ML4 Osmotic Pumps (ALZET, Cupertino, CA, USA) filled with 300 mM phospho-ICG-001 (a water-soluble version of the compound) solubilized in phosphate buffered saline (PBS) or PBS vehicle control were primed overnight at 37 °C and subsequently implanted subcutaneously. Every other day, the pumps were mobilized manually under the skin of the mice to prevent adhesions. 

### 4.5. Quantitative Reverse Transcription Polymerase Chain Reaction (RT-qPCR)

Samples were stored in TRIzol reagent (Thermo Fisher Scientific) at −80 °C. Total RNA extraction was performed according to the manufacturer’s protocol. RNA quality and quantity was measured using a NanoDrop 2000 (NanoDrop Products, Wilmington, DE, USA). Samples were considered good quality with A260/280 ratio > 1.8. 

For first strand synthesis 1 μg of total RNA and the qScript cDNA Supermix (Quantabio, Beverly, MA, USA) were used as follows: 25 °C for 5 min, 42 °C for 30 min, 85 °C for 5 min (T100 thermal cycler, Bio-Rad, Irvine, CA, USA). For qPCR, a SYBR Green RT-qPCR master mix was used (Quantabio, Beverly, MA, USA) combining: 2 μL cDNA, 12.5 μL SYBR Green master mix, 1 μL each forward and reverse primer probes, 8.5 μL DNase and RNase free deionized water. The reaction settings were: 95 °C for 3 min, 40 cycles of 95 °C for 20 s, 60 °C for 20 s 72 °C for 30 s (CFX96 Real-Time system, Bio-Rad). Primer sequences were obtained via the Harvard primer bank and synthesized by ValueGene (San Diego, CA, USA). All primer sequences are listed in Appendix A. 

### 4.6. RNA Seq Library Preparation and Data Analysis

MDA-MB-231 samples were prepared in biological duplicates after treatment for 48 h with 10 μM ICG-001 or DMSO vehicle control. The Ovation RNA-Seq System V2 and Ovation Ultralow Library System V2 (NuGEN Technologies, Inc., San Carlos, CA, USA) were used for amplification of 100 ng of total RNA and library prep. The cDNA was sheared using a Covaris S220 Focused-ultrasonicator (Covaris, Woburn, MA, USA). For quality control, the fragmentation product and libraries were validated using a 2100 Bioanalyzer Instrument (Agilent Technologies, Santa Clara, CA, USA). Whole transcriptome RNA Seq was performed on an Illumina HiSeq 2000 (Illumina, San Diego, CA, USA) using paired-end sequencing yielding 40–60 × coverage per sample. 

FastQ files were analyzed using Partek Flow software (Partek Inc., Chesterfield, MO, USA). Reads with a Phred quality score of 30 or higher were aligned to the International Human Genome Sequencing Consortium human reference genome (hg) 19 using the Spliced Transcripts Alignment to a Reference (STAR) aligner. The aligned reads were annotated and quantified using the Ensembl gene annotation database. From this feature list a differentially expressed gene list was created (false discovery rate (FDR) adjusted *p* < 0.05 ≥ |2| fold expression change), comparing MDA-MB-231 treated with ICG-001 to DMSO vehicle control. Gene ontology and pathway analysis was performed using Ingenuity Pathway Analysis (IPA) (Qiagen, Hilden, Germany).

### 4.7. Public Breast Cancer Gene Expression Data Repositories

cBioPortal was used to analyze FOXM1 target gene expression and reverse phase protein array (RPPA) data in The Cancer Genome Atlas (TCGA) breast cancer data set (*n* = 817). The TNBC subset included 82 cases and was analyzed separately. TCGA overall survival (OS) data was adjusted for 5-year OS and XLSTAT was used to perform Kaplan-Meier survival analysis. The University of California, Santa Cruz (UCSC) Cancer Genome Browser was used to evaluate FOXM1 expression in BC subtypes. Oncomine was used to query the TCGA BC dataset and METABRIC data set for CBP and FOXM1 expression.

### 4.8. Fluorescence Activated Cell Sorting (FACS) and Side Population Assay

Cells were dispersed with non-enzymatic cell dissociation solution (Sigma Aldrich, St. Louis, MO, USA) to avoid antigen digestion and re-suspended in FACS buffer (PBS + 2% FBS). 1 × 10^6^ cells per 100 μL were incubated for 15 min on ice with 10 μL of allophycocyanin (APC)-conjugated mouse anti-human CD44 antibody (G44-26) and fluorescein isothiocyanate (FITC)-conjugated mouse anti-human CD24 antibody (ML5) (both BD Biosciences). For dead cell exclusion, 4′, 6-diamidino-2-phenylindole (DAPI) (Sigma-Aldrich, St. Louis, MO, USA) nuclear dye solution was added. FACS was performed on a BD LSRFortessa (BD Biosciences).

The side population assay was used to quantify CSC cells [44]. Samples (1 × 10^6^ cells per mL) were incubated with 5 μg of Hoechst 33342 dye (Sigma Aldrich) for 2 h at 37 °C in a cell culture incubator with gentle mixing every 30 min. For negative controls, either 100 μM verapamil (Sigma-Aldrich) or 10 μM fumitremorgin C (FTC) (Sigma-Aldrich) was added. The samples were immediately placed on ice and washed with ice cold PBS. Propidium iodide (PI) (5 μg/mL) was added for dead cell exclusion. Samples were analyzed using a BD LSRFortessa (BD Biosciences). The side population gate was established using negative controls.

### 4.9. Protein Quantification

Samples were harvested into ice-cold PBS plus protease inhibitor (Merck Millipore, Billerica, MA, USA) and 1 μM Dithiothreitol (DTT) (Sigma-Aldrich) and stored at −80 °C until further use. For protein extraction, the Pierce NE-PER Nuclear and Cytoplasmic Extraction Kit (Thermo Fisher Scientific) was used. Protein quantification was performed using a Bio-Rad Protein Assay (Bio-Rad) and a SpectraMax M3 spectrophotometer (Molecular Devices, Sunnyvale, CA, USA).

Protein extract was mixed 1:1 with Laemmli Sample Buffer (Bio-Rad) containing 5% of 2-mercaptoethanol (Sigma-Aldrich, St. Louis, MO, USA) and boiled for 5min. Proteins were separated using pre-cast 4–20% gradient gels (PAGEr Gold Precast gels, Lonza, Basel, Switzerland) and transferred onto nitrocellulose overnight at 4 °C. Membranes were blocked for 60 min (5% skim milk) and incubated with primary antibody in blocking solution over night at 4 °C. Primary antibodies: mouse anti-human β-catenin (610153, BD Biosciences), mouse anti-human FOXM1 (A-11, Santa Cruz Biotechnologies-SCBT, Dallas, TX, USA), rabbit anti-human CBP (A-22, Santa Cruz Biotechnologies), mouse anti-human survivin (NB-500-205, Novus Biologicals, LLC, Littleton, CO, USA), mouse anti-human ACTB (H-102, SCBT) mouse anti-human GAPDH (6C5, SCBT) mouse anti-human ABCG2 (MAB4155, Millipore, Billerica, MA, USA), α-tubulin (B-5-1-2, SCBT) and mouse anti-human Lamin A/C (346, SCBT). Secondary antibody incubation was performed for 60 min at RT: goat anti-mouse IgG HRP conjugated, goat anti-rabbit IgG HRP conjugated (all from SCBT). Protein bands were visualized using HRP substrate (GE Healthcare Life Sciences, Pittsburgh, PA, USA) and a ChemiDoc MP gel imaging system (Bio-Rad). ImageJ software was used for digital quantification.

### 4.10. Co-Immunoprecipitation (CoIP)

250 μg of nuclear lysate was used per antibody and treatment condition for IP. 2 μg of pull-down antibody were added per sample and incubated at 4 °C overnight on a nutating mixer (VWR). The antibodies used were: CBP (A-22), FOXM1 (A-1) and normal rabbit IgG (all SCBT). Antibody bound proteins were immobilized using Protein A Resin beads (GBioscience, St. Louis, MO, USA) for 1 h at 4 °C und constant nutation. After several washing steps, the protein-bound beads were boiled in 30 μL Laemmli sample buffer (Bio-Rad) for 5 min and used for western blotting. 

### 4.11. Tissue Microarrays (TMAs)

TMAs were acquired from the Mid-Atlantic Division of the Cooperative Human Tissue Network-based at the University of Virginia containing a total of 398 stage II invasive BC cores, as well as 20 normal breast tissue cores, 12 breast fibroadenoma cores, 5 breast cell line controls (MCF-7, MCF-10a, SK-BR-3, SK-OV-3, T-47D) and 5 non-breast controls (salivary gland, endometrium, colon, tonsil, prostate). After exclusion of damaged sections and cases with missing information, 316 cases were included. The following optimized staining conditions were chosen: monoclonal mouse anti-human FOXM1 (A-11, SCBT) at 1:120 dilution, and monoclonal mouse anti-human CBP (C-20, SCBT) at 1:100. Staining was performed on 4 slides per antibody using a BOND-III Automated IHC/ISH Stainer (Leica, Buffalo Grove, IL, USA) by the USC Clinical Immunohistochemistry laboratory. A three-tier scoring system was devised: 0-no staining, 1-weakly positive, 2-strongly positive. Only tumors with a score 2 were included in the subsequent analysis. Clinical variables used for TMA analysis are listed in Appendix A.

### 4.12. Statistical Analysis

GraphPad PRISM (GraphPad Software Inc., La Jolla, CA, USA) and XLSTAT (Addinsoft, New York, NY, USA) were used for statistical analysis. For experiments with single variables and groups of two the Mann-Whitney test was used. For three or more groups, non-parametric one-way ANOVA (Kruskal-Wallis and Dunn’s multiple comparison) was used. The Sidak test was used to correct for multiple comparisons. A multivariate logistic regression model was used to calculate odds ratios. Cox regression analysis was used to determine hazard ratios. Kaplan-Meier analysis was used to generate survival curves. A two-tailed *p*-value of 0.05 was considered significant. Statistically insignificant results are listed n.s.

## 5. Conclusions

In conclusion, the current study describes CBP/β-catenin/FOXM1 as a novel molecular driver of TNBC biology. In vitro and in vivo studies demonstrated the benefit of targeting CBP/β-catenin/FOXM1 via ICG-001 by targeting drug resistant CSCs, which could provide a rationale for potential clinical trials in TNBC.

## Figures and Tables

**Figure 1 cancers-10-00525-f001:**
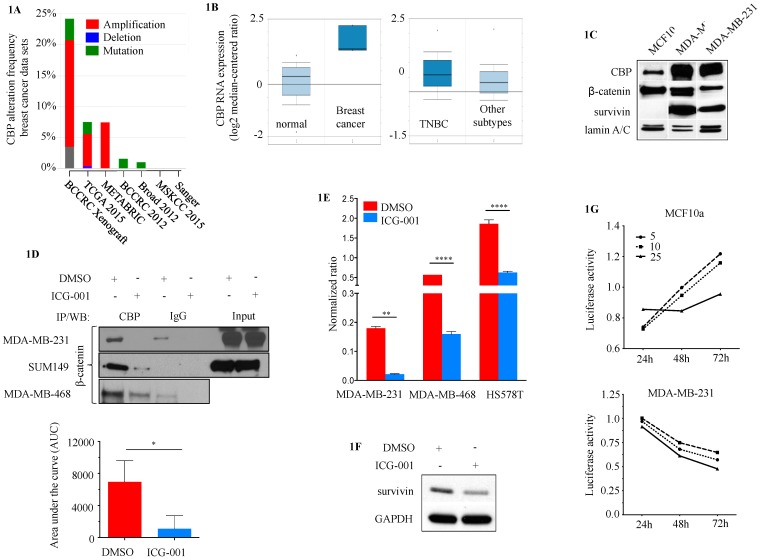
CBP as a potential target in TNBC. (**A**) Seven publicly available data sets showed genetic alterations in CBP in breast cancer (cBioPortal). (**B**) RNA expression levels of CBP in the TCGA BC data set (*n* = 593); left box plot: normal breast tissue compared to BC (2.91-fold BC vs. normal, *p* = 0.015), right box plot TNBC compared to BC other subtypes (1.18-fold TNBC vs. others, *p* = 0.012) (Oncomine database). (**C**) CBP, survivin and β-catenin protein levels in two TNBC cell lines (MDA-MB-231, MDA-MB-468) and non-tumorigenic epithelial breast cell line MCF10a. (**D**) Co-Immunoprecipitation (CoIP) of CBP/β-catenin in three TNBC cell lines (MDA231, MDA468 and SUM149) under DMSO vehicle control conditions and after treatment with 20 μM ICG-001 for 24 h (DMSO 6961 ± 2647 vs. ICG-001 1093 ± 1640). The area under the curve (AUC) refers to summary results for MDA-MB-231, MDA-MB-468 and SUM149 for CBP/b-catenin binding under DMSO (red bar) and ICG-001 (blue bar) treatment conditions. (**E**) Survivin-promoter driven luciferase reporter activity in three TNBC cell lines (MDA-MB-231, MDA-MB-468 and Hs578T) treated for 24 h with 10 μM ICG-001 or DMSO vehicle control. (**F**) Western blot for survivin expression MDA-MB-231 treated for 24 h with 10 μM ICG-001 or DMSO vehicle control. (* *p* < 0.05, ** *p* < 0.01, **** *p* < 0.0001). (**G**) Cell viability of not non-transformed MCF10a cells (top panel) and MDA-MD-231 TNBC cells (bottom panel) treated for up to 72 h with different concentrations of ICG-001.

**Figure 2 cancers-10-00525-f002:**
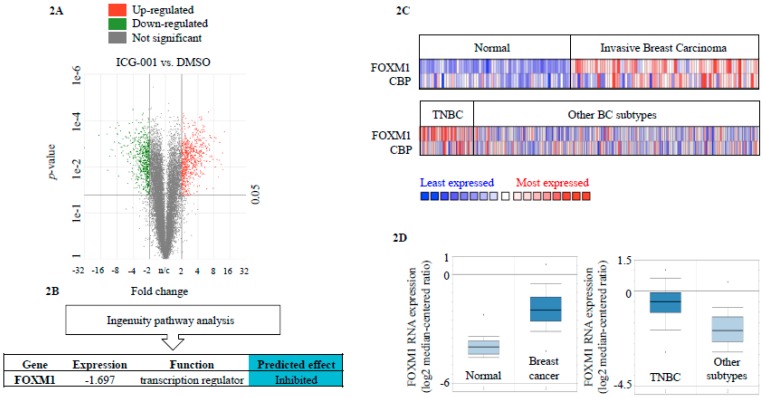
Chemical-genomic approach identifies FOXM1 as a downstream effector of CBP signaling in TNBC. (**A**) Whole transcriptome RNA Seq data volcano plot of differentially expressed genes in MDA-MB-231 treated with 10 μM ICG-001 or DMSO vehicle control (DMSO vs. ICG-001: 1339 differentially expressed genes, 729 genes up-regulated, 610 genes down-regulated FDR ≤ 0.05, ≤ |2-fold| change). (**B**) Ingenuity pathway analysis of RNA Seq data differential gene expression data identifies FOXM1 as an upstream regulatory factor. (**C**) Comparison of CBP and FOXM1 in the TCGA BC data set (*n* = 593); top panel: Normal and invasive breast cancer, bottom panel: TNBC and other BC subtypes. Colors are standard scores (*z*-scores) to depict relative normalized values within rows and do not allow for comparison of values between rows. (**D**) RNA expression levels of FOXM1 in the TCGA BC data set (*n* = 593); left box plot: normal breast tissue compared to BC (4.23-fold BC vs. normal, *p* = 3.51 × 10^−30^), right box plot: TNBC compared to other BC subtypes (2.43-fold TNBC vs. others, *p* = 1.24 × 10^−13^) (Oncomine database).

**Figure 3 cancers-10-00525-f003:**
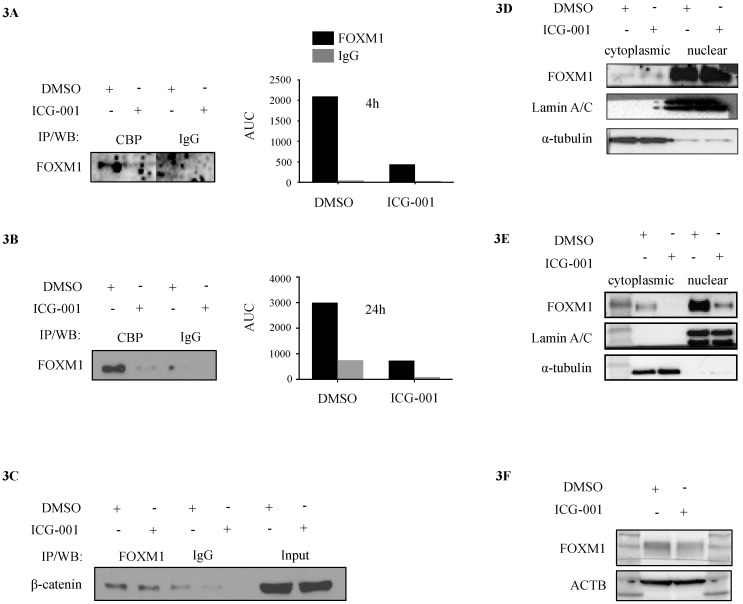
Effect of ICG-001 on CBP/FOXM1/β-catenin binding and FOXM1 protein levels. (**A**) CoIP of CBP with FOXM1 in MDA-MB-231 treated with for 4h and (**B**) 24 h with 20 μM ICG-001 or DMSO vehicle control (AUC DMSO 6006 ± 1500 vs. ICG-001 1480 ± 185, *n* = 1) (IP–Immunoprecipitation, WB–Western blot). (**C**) FOXM1 pull down and staining for associated β-catenin with or without addition of ICG-001. (**D**) Western blot showing protein levels of FOXM1 after 4 h and (**E**) 24 h treatment of MDA-MB-231 with ICG-001 or DMSO (cytosolic fraction in columns 2 and 3, nuclear fraction in columns 4 and 5). The first column is a protein marker to determine the molecular weight of each protein band. (**F**) FOXM1 levels in MDA-MB-468 TNBC cells after 24 h treatment with ICG-001 or DMSO control. The first and last column is a protein marker to determine the molecular weight of each protein band.

**Figure 4 cancers-10-00525-f004:**
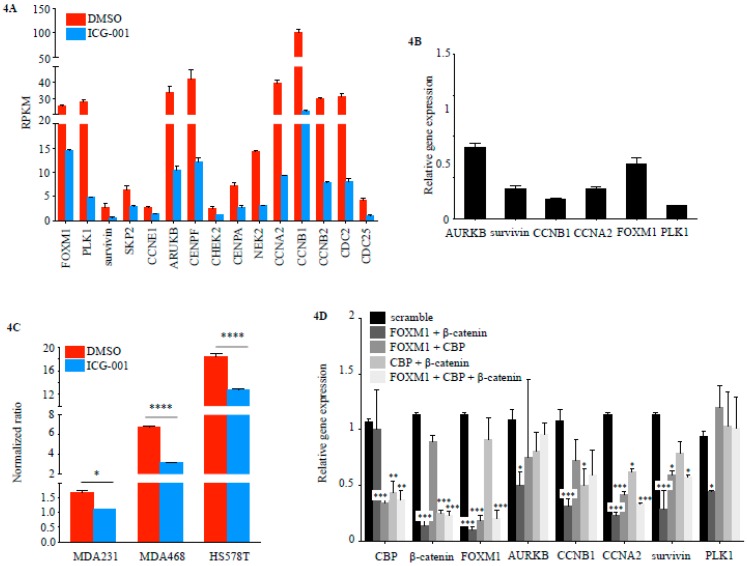
ICG-001 treatment effect FOXM1 transcriptional activity in TNBC cell lines. (**A**) RNA Seq. differential gene expression (RPKM) in FOXM1 target genes in MDA-MB-231 cells treated for 48 h with either 10 μM ICG-001 or DMSO vehicle control (ICG-001 vs. DMSO, *n* = 2 per time point per condition). (**B**) Expression changes (RT-qPCR) in FOXM1 target genes in MDA-MB-231 cells treated for 24 h with either 10 μM ICG-001 or DMSO vehicle control (*n* = 3 per condition). (**C**) FOXM1luc^Firefly/Renilla^ reporter for three TNBC cell lines (MDA-MB-231, MDA-MB-468, Hs578T) treated for 24 h with either 10 μM ICG-001 or DMSO vehicle control (bars represent normalized ratios of FOXM1 Firefly luciferase to control vector Renilla luciferase expression; *n* = 3 per condition per cell line). (* *p* < 0.05, **** *p* < 0.0001). (**D**) Transient siRNA gene knockdown (KD) (48 h) in MDA-MB-231 cells and qPCR for FOXM1 target. (* *p* < 0.05, ** *p* < 0.01, *** *p* < 0.001).

**Figure 5 cancers-10-00525-f005:**
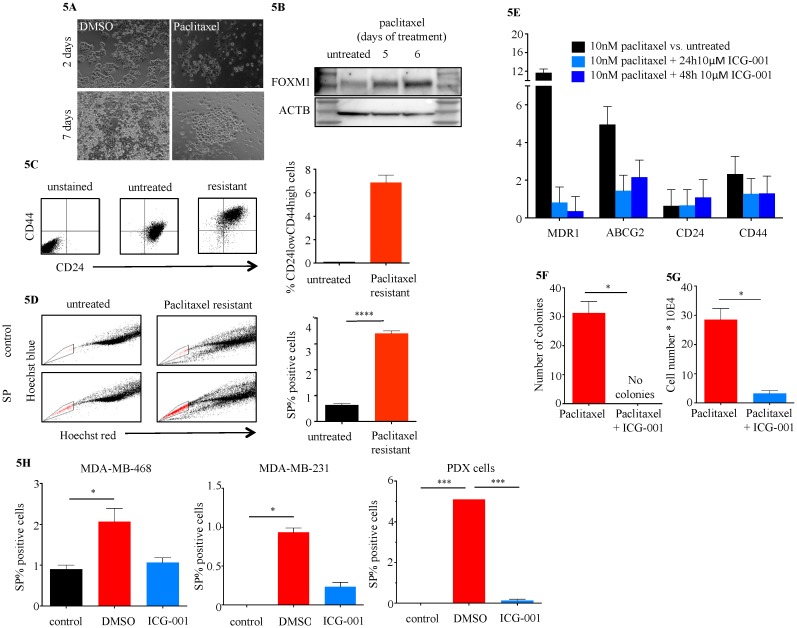
Paclitaxel treatment increases FOXM1 and CSC like cell populations and can be ameliorated with ICG-001. (**A**) Paclitaxel resistant MDA-MB-468 cell colony outgrowth after 7 d treatment with 10 nM Paclitaxel. (**B**) FOXM1 protein levels in Paclitaxel resistant MDA-MB-468 compared to treatment naive cells. (**C**) CSC like CD44^High^CD24^Low^ cells in naïve vs. Paclitaxel resistant MDA-MB-468 cells (*p* < 0.0001). (**D**) Side population (SP) cells in naïve vs. Paclitaxel resistant MDA-MB-468 cells (untreated 0.63 ± 0.03% vs. resistant 3.4 ± 0.06). (**E**) CSC marker gene expression in paclitaxel resistant MDA-MB-468 (post 6 d treatment) treated with for 24 h (light blue) or 48 h (dark blue) with ICG-001 or DMSO control (*n* = 3 per condition per time point). (**F**) Effect of treatment with 10 nM Paclitaxel plus 10 μM ICG-001 on resistant cell outgrowth compared to Paclitaxel only (*n* = 3 per condition). (**F**) Effect of re-treatment of Paclitaxel resistant MDA-MB-468 (post 5d Paclitaxel) with Paclitaxel only or Paclitaxel plus ICG-001 (*n* = 3 per condition). (**H**) Effect of 24 h 10 μM ICG-001 on SP cells in MDA-MB-468, MDA-MB-231 and patient derived xenograft (PDX) TNBC cells (*n* = 3 per cell model and condition) (control values for MDA-MB-231 and PDX-derived cells were 0%) (* *p* < 0.05, *** *p* < 0.001, **** *p* < 0.0001).

**Figure 6 cancers-10-00525-f006:**
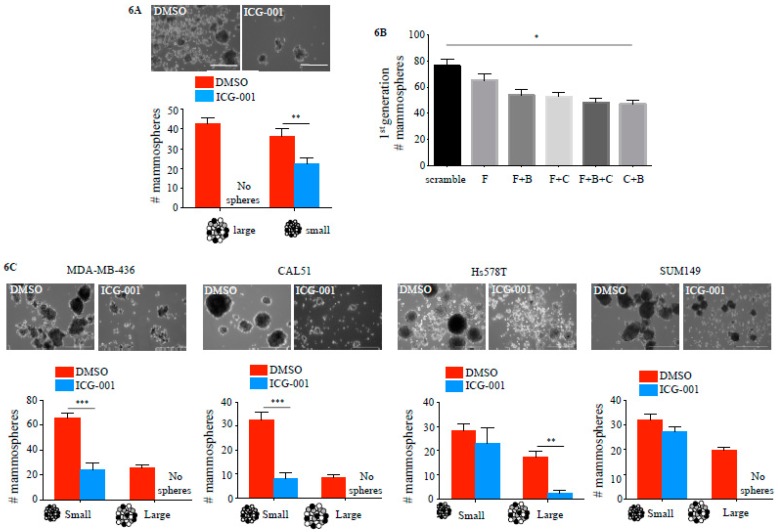
Targeting CBP reduces tumor initiation capacity in TNBC cell lines. (**A**) Tumorsphere formation of MDA-MB-231 cells treated for 9 days with 10 μM ICG-001 or DMSO control (20,000 cells per well, *n* = 3 per condition) (scale bar = 100 μm). (**B**) Seven-day culture of MDA-MB-231 after 48 h siRNA KD (F–FOXM1, B–β-catenin, C–CBP) (20,000 cells per well, *n* = 3 per condition). (**C**) Tumorsphere formation in additional TNBC cell lines (MDA-MB-436, CAL51, Hs578T and SUM149) after 7 days in culture, treated with 10 μM ICG-001 or DMSO vehicle control (*n* = 3 per cell line per condition). Large spheres < 50 μm diameter, small spheres > 50 μm (scale bar = 100 μm) (* *p* < 0.05, ** *p* < 0.01, *** *p* < 0.001).

**Figure 7 cancers-10-00525-f007:**
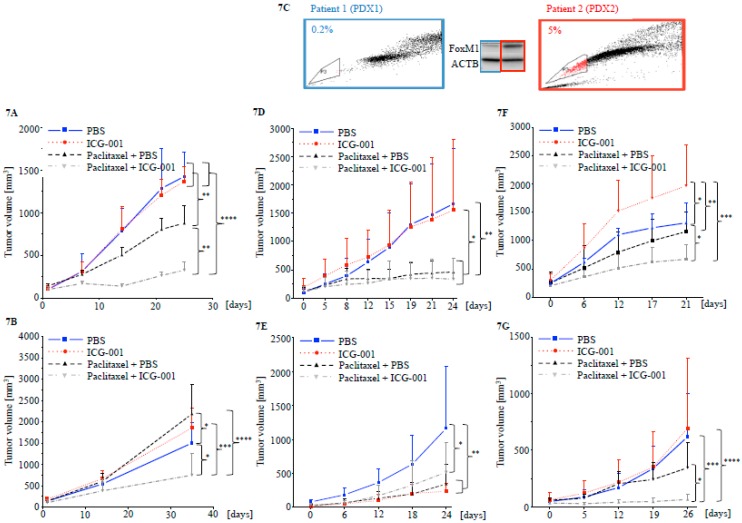
ICG-001 sensitizes TNBC cell line xenografts to Paclitaxel treatment and reduces tumor initiation capacity. (**A**) MDA-MB-468 TNBC cell xenograft in female Nod/SCID/gamma (NSG) mice treated for 26 days (*n* = 4 mice per treatment condition). Results were statistically significant for PBS vs. Paclitaxel + PBS, ICG-001 vs. Paclitaxel + PBS, PBS vs. Paclitaxel + ICG-001, ICG-001 vs. Paclitaxel + ICG-001 and PBS vs. Paclitaxel + PBS vs. Paclitaxel + ICG-001. (**B**) Secondary implantation of MDA-MB-468 tumors from primary xenografts. Tumors were dissociated after termination of the experiment and implanted into new female NSG mice without further treatment (*n* = 5 mice per treatment condition). Results were statistically significant for PBS vs. Paclitaxel + PBS, ICG-001 vs. Paclitaxel + PBS, PBS vs. Paclitaxel + ICG-001, ICG-001 vs. Paclitaxel + ICG-001 and PBS vs. Paclitaxel + PBS vs. Paclitaxel + ICG-001. (**C**) FOXM1 protein expression and proportion of side population cells in tumors from two TNBC patients. (**D**,**E**) PDX1 tumor growth and treatment response after primary implantation (results were statistically significant for PBS vs. paclitaxel + PBS, PBS vs. paclitaxel + ICG-001, ICG-001 vs. paclitaxel + PBS and ICG-001 vs. paclitaxel + ICG-001) (**D**) and secondary implant without further treatment (results were significant for PBS vs. ICG-001, PBS vs. paclitaxel + PBS and PBS vs. paclitaxel + ICG-001) (**E**). (**F** and **G**) PDX2 tumor growth and treatment response after primary implantation (results were statistically significant for PBS vs. ICG-001, PBS vs. paclitaxel + ICG-001, ICG-001 vs. paclitaxel + PBS and ICG-001 vs. paclitaxel + ICG-001) (**F**) and secondary implant without further treatment (results were statistically significant for PBS vs. paclitaxel + ICG-001, ICG-001 vs. paclitaxel + ICG-001 and ICG-001 vs. paclitaxel + PBS) (**G**). (* *p* < 0.05, ** *p* < 0.01, *** *p* < 0.001, **** *p* < 0.0001).

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
