# Peer review of "CBP/β-Catenin/FOXM1 Is a Novel Therapeutic Target in Triple Negative Breast Cancer"

_cancers, 2018, doi:10.3390/cancers10120525_

Reviewer 1 Report

CBP/β-catenin/FOXM1 is a novel therapeutic target in triple negative breast cancer

This article is written well, but lacks in new knowledge and grounds. Therefore, I require Major Revision.

Major point

Please describe the analysis method of this study in detail in material and methods section.

Figures are not unclear, replacement is necessary. Analysis is necessary again (Figure 2).

The reproducibility of the experiment is not clear. Please prove this.

To prove this conclusion, data is scarce. Please prove by additional experiment.

(exc, Immunohistochemical staining using clinical specimens.)

And, please describe the discussion in more detail.

Minor point

The sentence of this paper has many careful mention errors. Please review it.

Author Response

We would like to thank the reviewer for evaluating our manuscript and provide constructive criticism and suggestions for improving the quality of the study. We are have answered the concerns of reviewer 2 and 3 and believe that the concerns raised by reviewer 1 have mostly been addressed with our correction. We provide our answers in the separate response sheets along with a corrected version of the manuscript.

Major point

Please describe the analysis method of this study in detail in material and methods section.

Figures are not unclear, replacement is necessary. Analysis is necessary again (Figure 2).

The reproducibility of the experiment is not clear. Please prove this.

To prove this conclusion, data is scarce. Please prove by additional experiment.

(exc, Immunohistochemical staining using clinical specimens.)

And, please describe the discussion in more detail.

Minor point

The sentence of this paper has many careful mention errors. Please review it.

Reviewer 2     

We would like to thank the reviewer for evaluating our manuscript and provide constructive criticism and suggestions for improving the quality of the study. We are happy to address the concerns below.

1. What was the rational for selecting 10-20 μM as the concentration of ICG-001 in different experiments? There was no data in the manuscript demonstrating the profile of cytotoxicity of ICG-001 in TNBC cell lines.

The initial concentration of ICG-001 was chosen at 20 μM for the Co-IP experiments as established previously by the Kahn lab to maximize the effect on CoIP to visualize the disruption of binding of CBP to β-catenin or FOXM1. We previously showed that ICG-001 effects gene expression in a dose dependent but consistent manner [1]. This paper also demonstrated that 10uM already leads to disruption of CBP/β-catenin binding, and that treatment with ICG-001 does not decrease protein levels [1].

Regarding cytotoxicity, we would like to emphasize that ICG-001 is not considered a cell-killing agent. Rather, the proposed mechanism of action is that of a differentiating agent targeting cancer cells by modulating their phenotype, e.g. changing the CSC phenotype and/or rendering cancer cells less resistant to chemotherapy. Hence, the results in our study are primarily shown as a combination therapy of ICG-001 with paclitaxel. Nevertheless, the reviewer has pointe out an important issue regarding the effect on ICG-001 on cancer cell viability. Although not intended as a single agent, ICG-001 has some effect on CBP-dependent MDA-MB-231 cells, while not affecting MCF10a cells. We included a supplementary figure showing the reduction in viable MDA-MB-231 cells but not MCF10a cells (Figure 1G).

2. Figure 1A and Figure S1A: Only 7 data sets are listed; not 10 as mentioned in the Figure legend.

We corrected this mistake the correct number is 7 studies and has been changed in the figure legend.

3. Figure 1B: Why was TCGA data set chosen among other data sets (shown in Figure 1A) for extracting information regarding the CBP RNA expression?

We used Oncomine to find breast cancer data sets with annotation for hormone receptor expression and RNA expression of CBP. The TCGA data set is the larger available which meets our criteria. We have now included the data set by Curtis et al [2], which shows similar results.

4. Figure 1C: It is of low quality. Also please clarify if samples were extracted from the whole cell lysate or nuclear fractionation was done on them.

The samples were fractionated and nuclear lysates were used for quantification of protein expression shown in Figure 1C.

5. Figure 1D: Which experiment does the graph showing the area under the curve refer to?

Area under the curve here refers to the analysis to quantify protein level using the Image J software. The software creates a curve that represents the amount of pixels in the plot, with more pixels corresponding to larger amount of protein. Quantifying the area under the curve can be used to quantify protein.

6. Figure 1F: Why was the inhibitory effect of ICG-001 on the expression of survivin only shown in MDA-MB-231 cells and not the other TNBC cell lines?

Figure 1E shows survivin reporter activity in three triple negative breast cancer cell lines and the inhibitory effect of treatment with ICG-001 on reporter activity. In figure 1F MDA-231 was chosen as a representative example for the effect of ICG-001 on survivin protein expression.

7. Line 79: Please add the value for the fold difference of FOXM1.

The fold change value has been added.

8. Figure 2: The labeling/numbering in Figure 2 (2 A-C) does not match the Figure legend (2 A-E) or the description in the main text (2 A-E).

We thank the reviewer noticing the labeling error. The figure legend and main text have been edited to match the data shown in the figure.

9. Figure S2: No arrows are to be found to indicate the differential gene expression before and after treatment with ICG-001 in MDA-MB-231 cells. 

Figure S2 shows genes in the FOXM1 transcriptional network that are overexpressed in TNBC cancer. The arrows mark genes that were found to be down regulated in MDA-231 treated with ICG-001. These results are shown in figure 4A. A clarifying statement has been added to the figure legend for figure S2

10. Figure 3C: Why is there no lamina/c in the drug treated group for cytosolic fraction?

The second and third column represents cytoplasmic extracts and a-tubulin was used as the housekeeping control. The third and fourth column represents nuclear extracts and lamin a/c was used as the housekeeping control. The first column is a protein marker. We agree that this presentation is confusing and added an explanatory statement to the figure legend.

11. There is a discrepancy in fold difference of reporter activity between untreated and ICG-001 treated MDA-MB-231 cells as shown in Figures 4C and S3B (6 fold and 1.5 fold, respectively). Which one is correct?

Thank you for pointing out this discrepancy. Two different reporter-constructs have been used for this analysis. The assay used in Figure 4C did not include a Renilla luciferase control, while the assay shown in figure S3B included these controls. We chose to include only the reporter results including the Renilla luciferase control as shown in figure S3B. We substituted this figure for figure 4C and edited the figure legends and main text accordingly.

12. Figure 5: The labeling/numbering in Figure 5 does not match the Figure legend or the description in the main text. Additionally, Figure 5G is missing columns for control groups of MDA-MB-231 and PDX cells. Figures 5C and 5D are of poor quality.

We corrected the figure to match the figure legend and main text. Regarding figure 5G, the values for the control were 0% in MDA-231 and PDX cells, and hence no bar is shown in the graph. A clarifying comment has been added to the figure legend. We have exchanged the figures in 5C and 5D with better quality images.

13. Line 139: Please clarify if the cells continuously treated with ICG-001 were the paclitaxel-resistant ones and whether they were simultaneously exposed to paclitaxel or after day nine, paclitaxel treatment was stopped and then continuous ICG-001 treatment was initiated.

We regret that this was not stated clearly in the original submission. The cells were treatment naïve and have not been treated with ICG-001.

14. Line 149: No data is found in Figure 6 regarding the second generation of mammospheres.

The error has been corrected.

15. Figure 6B: Shouldn’t one expect to have the lowest number of mammospheres in F+B group based on data presented in Figure 4D?

It is true that combined knock-down of ICG-001 had the strongest effect on gene-expression of the genes investigated. The fact that CBP plus b-catenin knockdown had the strongest effect on mammosphere formation could be due to further changes not captured by the current assay. This could be investigated in future studies. A statement regarding this issue is already included in the discussion section been added to the discussion section line 266-271).

16. Line 196: There is no Table 3 in the main text.

The error has been corrected.

17. Line 158: Why did the tumors in mice bearing PDX1 showed more aggressive growth (Figure S4A) and worse response to therapy (as compared to mice bearing PDX2), while PDX1 was from a tumor with low FOXM1 protein expression and a small fraction of side population cells (0.2%), and PDX2 was from a tumor with relatively higher expression of FOXM1 and higher proportion of side population cells (5%)?

This is a labeling error that also applies to comment 19. The error has been corrected.

18. Figure 4S: The labeling does not match the Figure legend.

The error has been corrected.

19. Line 165: “Quantification of FOXM1 and ABCG2 in PDX1 bearing mice post treatment showed reduced protein expression after treatment with paclitaxel plus ICG-001 (Supplementary figure S4B and S4C). These data suggest that targeting CBP/FOXM1 via the small molecule inhibitor ICG-001 enhanced the initial response and duration of remission after paclitaxel chemotherapy in vivo”. However, authors mentioned in line 159 that “PDX1 tumors responded similarly to paclitaxel alone or combination of paclitaxel plus ICG-001 (Figure 7D and supplementary table S9)”. Which claim is correct?

There was a labeling error. The wording in the main text should have been “Quantification of FOXM1 and ABCG2 in PDX2 bearing mice”. PDX2 is the tumor with high FOXM1 expression. This data makes the point that tumors with high FOXM1 expression respond poorly to paclitaxel, potentially due to a higher degree of CSC like cells and drug resistant phenotype, exemplified here by the expression of the CSC and drug resistance marker ABCG2. Treatment with ICG-001 reduces the level of both FOXM1 and ABCG2 in vivo (and also in vitro as shown in figures 3C, 4A as well as 5E, respectively) and increases the cell killing efficiency of paclitaxel.

20. Figure 7B: Why does tumour volume significantly increase in the group treated with paclitaxel plus PBS as compared to one treated with PBS alone?

Our explanation for this is that the PDX tumors pre-treated with paclitaxel are enriched for CSC or tumor initiating cells (similar to what we show in figure 5D where paclitaxel treatment increases the number of SP cells). Hence, the engraftment and outgrowth in the second-generation xenografts would be more efficient, resulting in larger tumors.

21. Line 217: Table S15 should be replaced with table S16.

The error has been corrected.

22. Line 218: Table S16 should be replaced with table S15.

The error has been corrected.

23. Line 223: “high FOXM1 expression in TNBC subtype”, it should be only “in TNBC subtype” according to table S18.

We would like to thank the reviewer for pointing out this mistake. The text has been changed accordingly.

24. Line 231: “Supplementary Figure S12C and S12D”. There are no such Figures in the supporting information. The authors are probably referring to Figure S7. The labeling in Figure S7 is incorrect as well.

The error has been corrected.

Reference:

1.         Emami, K.H., et al., A small molecule inhibitor of beta-catenin/CREB-binding protein transcription [corrected]. Proc Natl Acad Sci U S A, 2004. 101(34): p. 12682-7.

2.         Curtis, C., et al., The genomic and transcriptomic architecture of 2,000 breast tumours reveals novel subgroups. Nature, 2012. 486(7403): p. 346-52.

Reviewer 3

We would like to thank the reviewer for evaluating our manuscript and provide constructive criticism and suggestions for improving the quality of the study. We are happy to address the concerns below.

1. The authors used different cell line for different experiments. Some experiments were conducted with MDA-MB-231, while others were conducted with MDA-MB-468 cells. Some were conducted with both. The authors did not explain the choice of different cell line in various experiments.

We used several triple TNBC cell lines in order to make the conclusions more generalizable. The in the case of CoIP experiments, we showed that treatment with ICG-001 reduces CBP/b-catenin binding in MDA-231, MBA-468 and SUM149. For CBP/FOMX1 binding we only validated the experiment in MDA-231 since we also had RNA-Seq data for this cell line. An effect of disrupting CBP/FOXM1-binding via ICG-001 was the reduction in FOXM1 protein levels, which we also show in MDA-468, albeit to a lesser degree.

2. All immunoprecipitation (IPs) studies were conducted 24 hours after treating the cells with ICG-001. Moreover, no input controls have been included. It is likely that ICG-001 inhibits the interaction of CBP and B-catenin by reducing the expression of either or both the proteins. Hence, instead of a reduction in interaction a loss of protein expression after 24 hours has been the contributing factor. Interactions can be observed as early as 1 hour. Hence, IP experiments need to be conducted for a short duration of treatment.

We agree with the observation made by the reviewer and have provided a 4h treatment time point showing the disruption of CBP/FOXM1 via ICG-001 in MDA-231 cells (new Figure 3A) after 4h and 24h. The data shows that protein levels of FOMX1 are not affected after a short treatment period, but the interaction with CBP is clearly reduced. We also have data showing that b-catenin levels are not affected. Although we didn’t look for changes in CBP protein levels, our lab has previously shown that treatment with ICG-001 does not affect CBP protein levels [1].

3. Although ICG-001 was discovered in 2004, its clinical track-record could not be determined. Alternatively, a new specific CBP inhibitor (Cell Centric’s CCS1477) has been advanced to the clinic for the treatment of prostate cancer. It is unknown at this point whether ICG-001 failed to show any tumor growth inhibition in animal models due to poor pharmacokinetic (PK) properties or lack of efficacy. No CBP-target genes were measured in the tumor specimens to determine the reason for the lack of efficacy. The experiments need to be validated with CCS1477, if available.

The clinical version of ICG-001 is PRI724, which has been is still in use in several clinical trials for Myeloid malignancies (NCT02828254), metastatic pancreatic carcinoma (NCT01764477), advanced solid tumors (NCT01302405). As we state below, ICG-001 is not considered to be a cytotoxic agent per se, but rather modulates the phenotype of cancer cells, specifically CSC-like cells as the current study as well as previous studies have shown (REF). There the viability studies have been conducted in combination with paclitaxel.

4. siRNA studies are conflicting. While FOXM1 and b-catenin siRNAs demonstrate strong inhibition of FOXM1 target gene expression, the effect of CBP siRNA was modest. One would expect that CBP, being downstream of B-catenin, should be the major player to affect the cellular properties.

This is an interesting point raised by the reviewer. A possible explanation for the observed results could be that gene- or protein dosage of FOXM1 and b-catenin are more critical than CBP, but the interaction of CBP with FOXM1 and b-catenin is critical. Hence, while siRNA knockdown cannot fully eliminate CBP from the cells, enough of the protein might be left to drive CBP/FOMX1/b-catenin dependent gene expression. On the other hand, treatment with ICG-001 potently disrupts the binding of CBP to the transcriptional complex, thereby inhibiting gene expression. An explanatory section has been added to the discussion of our manuscript.

5. Control cell lines such as non-cancerous lines have not been used to demonstrate selectivity of ICG-001 on the proliferation of TNBC lines.

This is an important point by the reviewer. We would like to emphasize that ICG-001 is not considered a cell-killing agent. Rather, the proposed mechanism of action is that of a differentiating agent targeting cancer cells by modulating their phenotype, e.g. changing the CSC phenotype and/or rendering cancer cells less resistant to chemotherapy. Hence, the results in our study are primarily shown as a combination therapy of ICG-001 with paclitaxel. Nevertheless, the reviewer has pointed out an important issue regarding the effect on ICG-001 on cancer cell viability. Although not intended as a single agent, ICG-001 has some, albeit a moderate effect on CBP-dependent MDA-MB-231 cells, while not affecting non-cancerous MCF10a cells with low CBP-levels (Figure 1C). We included a supplementary figure showing the reduction in viable MDA-MB-231 cells but not MCF10a cells (Figure 1G). Regarding the moderate effect on TNBC cell viability we would like to stress again that ICG-001 is not intended to be used a single agent.

6. The authors have not provided an explanation for the lack of efficacy of ICG-001 in vivo. Combination treatments were effective, while ICG-001 individual treatment failed to demonstrate any effect. It is unclear how ICG-001 worked in vitro, but not in vivo.

ICG-001 is not a cytotoxic agent per se and does not kill tumor cells directly. Instead the proposed function is that of a modulating or differentiating agent, which leads to a change in CSC phenotype and related drug resistance. We demonstrate this by showing in vitro that ICG-001 strongly reduces the SP-phenotype and CSC-related gene expression (such as MDR1, ABCG2 and CD44). The observation that ICG-001 inhibits cell growth in vitro but not in vivo might be due the fact that cell cultures in vitro provide only a basic environment for cell growth, while in vivo an complex environment may provide additional growth cues for the cells. The important part of our study is that treatment with ICG-001 sensitizes previously resistant cells to paclitaxel and also inhibits the occurrence of CSC-like cells, which are thought to be responsible for tumor initiation and regrowth after initially successful treatment. We show this in vitro (Figure 5G) as well as in vivo by the secondary implantation of pre-treated tumors (Figure 7G).

7. The authors have also failed to provide any information on the patient-derived xenografts (PDXs) and PDX-derived cells.

Supplementary table S8 gives an overview of the origin of the human primary breast tumor tissues used for the PDX models.

8. Overall, although the authors have attempted several models, the data did not drive home a clear conclusion on the importance of CBP in TNBC.

We hope that the addition of new data requested by the reviewer as well as our explanations to the various led to a clearer picture and established the role of CBP in TNBC

Reference:

1.         Emami, K.H., et al., A small molecule inhibitor of beta-catenin/CREB-binding protein transcription [corrected]. Proc Natl Acad Sci U S A, 2004. 101(34): p. 12682-7.

Reviewer 2 Report

In this study, the authors aimed to evaluate the effect of targeting of the CBP/β-catenin/FOXM1 signaling pathway with ICG-001 (candidate drug) on elimination of cancer stem cells (CSCs) and subsequently sensitization of TNBC tumors to chemotherapy (paclitaxel treatment). ICG-001 is a CBP/β-catenin antagonist that can block CBP/β-catenin-mediated transcription. The obtained results suggest that targeting of the CBP/β-catenin/FOXM1 signaling pathway with ICG-001 can sensitize the paclitaxel-resistant cells and enhance the therapeutic efficacy.

In general, the study was performed with an adequate design. However, many typos were found throughout the text. Many of the Figures were either mislabeled in the main text/supporting information or were just not to be found. Moreover, many of the Figures suffer from poor quality and could be further improved.

Some points to clarify:

1.       What was the rational for selecting 10-20 μM as the concentration of ICG-001 in different experiments? There was no data in the manuscript demonstrating the profile of cytotoxicity of ICG-001 in TNBC cell lines.

2.       Figure 1A and Figure S1A: Only 7 data sets are listed; not 10 as mentioned in the Figure legend.

3.       Figure 1B: Why was TCGA data set chosen among other data sets (shown in Figure 1A) for extracting information regarding the CBP RNA expression?

4.       Figure 1C: It is of low quality. Also please clarify if samples were extracted from the whole cell lysate or nuclear fractionation was done on them.

5.       Figure 1D: Which experiment does the graph showing the area under the curve refer to?

6.       Figure 1F: Why was the inhibitory effect of ICG-001 on the expression of survivin only shown in MDA-MB-231 cells and not the other TNBC cell lines?

7.       Line 79: Please add the value for the fold difference of FOXM1.

8.       Figure 2: The labeling/numbering in Figure 2 (2 A-C) does not match the Figure legend (2 A-E) or the description in the main text (2 A-E).

9.       Figure S2: No arrows are to be found to indicate the differential gene expression before and after treatment with ICG-001 in MDA-MB-231 cells. 

10.    Figure 3C: Why is there no lamina/c in the drug treated group for cytosolic fraction?

11.    There is a discrepancy in fold difference of reporter activity between untreated and ICG-001 treated MDA-MB-231 cells as shown in Figures 4C and S3B (6 fold and 1.5 fold, respectively). Which one is correct?

12.    Figure 5: The labeling/numbering in Figure 5 does not match the Figure legend or the description in the main text. Additionally, Figure 5G is missing columns for control groups of MDA-MB-231 and PDX cells. Figures 5C and 5D are of poor quality.

13.    Line 139: Please clarify if the cells continuously treated with ICG-001 were the paclitaxel-resistant ones and whether they were simultaneously exposed to paclitaxel or after day nine, paclitaxel treatment was stopped and then continuous ICG-001 treatment was initiated.

14.    Line 149: No data is found in Figure 6 regarding the second generation of mammospheres.

15.    Figure 6B: Shouldn’t one expect to have the lowest number of mammospheres in F+B group based on data presented in Figure 4D?

16.    Line 196: There is no Table 3 in the main text.

17.    Line 158: Why did the tumors in mice bearing PDX1 showed more aggressive growth (Figure S4A) and worse response to therapy (as compared to mice bearing PDX2), while PDX1 was from a tumor with low FOXM1 protein expression and a small fraction of side population cells (0.2%), and PDX2 was from a tumor with relatively higher expression of FOXM1 and higher proportion of side population cells (5%)?

18.    Figure 4S: The labeling does not match the Figure legend.

19.    Line 165: “Quantification of FOXM1 and ABCG2 in PDX1 bearing mice post treatment showed reduced protein expression after treatment with paclitaxel plus ICG-001 (Supplementary figure S4B and S4C). These data suggest that targeting CBP/FOXM1 via the small molecule inhibitor ICG-001 enhanced the initial response and duration of remission after paclitaxel chemotherapy in vivo”. However, authors mentioned in line 159 that “PDX1 tumors responded similarly to paclitaxel alone or combination of paclitaxel plus ICG-001 (Figure 7D and supplementary table S9)”. Which claim is correct?

20.    Figure 7B: Why does tumour volume significantly increase in the group treated with paclitaxel plus PBS as compared to one treated with PBS alone?

21.    Line 217: Table S15 should be replaced with table S16.

22.    Line 218: Table S16 should be replaced with table S15.

23.    Line 223: “high FOXM1 expression in TNBC subtype”, it should be only “in TNBC subtype” according to table S18.

24.    Line 231: “Supplementary Figure S12C and S12D”. There are no such Figures in the supporting information. The authors are probably referring to Figure S7. The labeling in Figure S7 is incorrect as well.

Author Response

Reviewer 2     

We would like to thank the reviewer for evaluating our manuscript and provide constructive criticism and suggestions for improving the quality of the study. We are happy to address the concerns below.

1. What was the rational for selecting 10-20 μM as the concentration of ICG-001 in different experiments? There was no data in the manuscript demonstrating the profile of cytotoxicity of ICG-001 in TNBC cell lines.

The initial concentration of ICG-001 was chosen at 20 μM for the Co-IP experiments as established previously by the Kahn lab to maximize the effect on CoIP to visualize the disruption of binding of CBP to β-catenin or FOXM1. We previously showed that ICG-001 effects gene expression in a dose dependent but consistent manner [1]. This paper also demonstrated that 10uM already leads to disruption of CBP/β-catenin binding, and that treatment with ICG-001 does not decrease protein levels [1].

Regarding cytotoxicity, we would like to emphasize that ICG-001 is not considered a cell-killing agent. Rather, the proposed mechanism of action is that of a differentiating agent targeting cancer cells by modulating their phenotype, e.g. changing the CSC phenotype and/or rendering cancer cells less resistant to chemotherapy. Hence, the results in our study are primarily shown as a combination therapy of ICG-001 with paclitaxel. Nevertheless, the reviewer has pointe out an important issue regarding the effect on ICG-001 on cancer cell viability. Although not intended as a single agent, ICG-001 has some effect on CBP-dependent MDA-MB-231 cells, while not affecting MCF10a cells. We included a supplementary figure showing the reduction in viable MDA-MB-231 cells but not MCF10a cells (Figure 1G).

2. Figure 1A and Figure S1A: Only 7 data sets are listed; not 10 as mentioned in the Figure legend.

We corrected this mistake the correct number is 7 studies and has been changed in the figure legend.

3. Figure 1B: Why was TCGA data set chosen among other data sets (shown in Figure 1A) for extracting information regarding the CBP RNA expression?

We used Oncomine to find breast cancer data sets with annotation for hormone receptor expression and RNA expression of CBP. The TCGA data set is the larger available which meets our criteria. We have now included the data set by Curtis et al [2], which shows similar results.

4. Figure 1C: It is of low quality. Also please clarify if samples were extracted from the whole cell lysate or nuclear fractionation was done on them.

The samples were fractionated and nuclear lysates were used for quantification of protein expression shown in Figure 1C.

5. Figure 1D: Which experiment does the graph showing the area under the curve refer to?

Area under the curve here refers to the analysis to quantify protein level using the Image J software. The software creates a curve that represents the amount of pixels in the plot, with more pixels corresponding to larger amount of protein. Quantifying the area under the curve can be used to quantify protein.

6. Figure 1F: Why was the inhibitory effect of ICG-001 on the expression of survivin only shown in MDA-MB-231 cells and not the other TNBC cell lines?

Figure 1E shows survivin reporter activity in three triple negative breast cancer cell lines and the inhibitory effect of treatment with ICG-001 on reporter activity. In figure 1F MDA-231 was chosen as a representative example for the effect of ICG-001 on survivin protein expression.

7. Line 79: Please add the value for the fold difference of FOXM1.

The fold change value has been added.

8. Figure 2: The labeling/numbering in Figure 2 (2 A-C) does not match the Figure legend (2 A-E) or the description in the main text (2 A-E).

We thank the reviewer noticing the labeling error. The figure legend and main text have been edited to match the data shown in the figure.

9. Figure S2: No arrows are to be found to indicate the differential gene expression before and after treatment with ICG-001 in MDA-MB-231 cells. 

Figure S2 shows genes in the FOXM1 transcriptional network that are overexpressed in TNBC cancer. The arrows mark genes that were found to be down regulated in MDA-231 treated with ICG-001. These results are shown in figure 4A. A clarifying statement has been added to the figure legend for figure S2

10. Figure 3C: Why is there no lamina/c in the drug treated group for cytosolic fraction?

The second and third column represents cytoplasmic extracts and a-tubulin was used as the housekeeping control. The third and fourth column represents nuclear extracts and lamin a/c was used as the housekeeping control. The first column is a protein marker. We agree that this presentation is confusing and added an explanatory statement to the figure legend.

11. There is a discrepancy in fold difference of reporter activity between untreated and ICG-001 treated MDA-MB-231 cells as shown in Figures 4C and S3B (6 fold and 1.5 fold, respectively). Which one is correct?

Thank you for pointing out this discrepancy. Two different reporter-constructs have been used for this analysis. The assay used in Figure 4C did not include a Renilla luciferase control, while the assay shown in figure S3B included these controls. We chose to include only the reporter results including the Renilla luciferase control as shown in figure S3B. We substituted this figure for figure 4C and edited the figure legends and main text accordingly.

12. Figure 5: The labeling/numbering in Figure 5 does not match the Figure legend or the description in the main text. Additionally, Figure 5G is missing columns for control groups of MDA-MB-231 and PDX cells. Figures 5C and 5D are of poor quality.

We corrected the figure to match the figure legend and main text. Regarding figure 5G, the values for the control were 0% in MDA-231 and PDX cells, and hence no bar is shown in the graph. A clarifying comment has been added to the figure legend. We have exchanged the figures in 5C and 5D with better quality images.

13. Line 139: Please clarify if the cells continuously treated with ICG-001 were the paclitaxel-resistant ones and whether they were simultaneously exposed to paclitaxel or after day nine, paclitaxel treatment was stopped and then continuous ICG-001 treatment was initiated.

We regret that this was not stated clearly in the original submission. The cells were treatment naïve and have not been treated with ICG-001.

14. Line 149: No data is found in Figure 6 regarding the second generation of mammospheres.

The error has been corrected.

15. Figure 6B: Shouldn’t one expect to have the lowest number of mammospheres in F+B group based on data presented in Figure 4D?

It is true that combined knock-down of ICG-001 had the strongest effect on gene-expression of the genes investigated. The fact that CBP plus b-catenin knockdown had the strongest effect on mammosphere formation could be due to further changes not captured by the current assay. This could be investigated in future studies. A statement regarding this issue is already included in the discussion section been added to the discussion section line 266-271).

16. Line 196: There is no Table 3 in the main text.

The error has been corrected.

17. Line 158: Why did the tumors in mice bearing PDX1 showed more aggressive growth (Figure S4A) and worse response to therapy (as compared to mice bearing PDX2), while PDX1 was from a tumor with low FOXM1 protein expression and a small fraction of side population cells (0.2%), and PDX2 was from a tumor with relatively higher expression of FOXM1 and higher proportion of side population cells (5%)?

This is a labeling error that also applies to comment 19. The error has been corrected.

18. Figure 4S: The labeling does not match the Figure legend.

The error has been corrected.

19. Line 165: “Quantification of FOXM1 and ABCG2 in PDX1 bearing mice post treatment showed reduced protein expression after treatment with paclitaxel plus ICG-001 (Supplementary figure S4B and S4C). These data suggest that targeting CBP/FOXM1 via the small molecule inhibitor ICG-001 enhanced the initial response and duration of remission after paclitaxel chemotherapy in vivo”. However, authors mentioned in line 159 that “PDX1 tumors responded similarly to paclitaxel alone or combination of paclitaxel plus ICG-001 (Figure 7D and supplementary table S9)”. Which claim is correct?

There was a labeling error. The wording in the main text should have been “Quantification of FOXM1 and ABCG2 in PDX2 bearing mice”. PDX2 is the tumor with high FOXM1 expression. This data makes the point that tumors with high FOXM1 expression respond poorly to paclitaxel, potentially due to a higher degree of CSC like cells and drug resistant phenotype, exemplified here by the expression of the CSC and drug resistance marker ABCG2. Treatment with ICG-001 reduces the level of both FOXM1 and ABCG2 in vivo (and also in vitro as shown in figures 3C, 4A as well as 5E, respectively) and increases the cell killing efficiency of paclitaxel.

20. Figure 7B: Why does tumour volume significantly increase in the group treated with paclitaxel plus PBS as compared to one treated with PBS alone?

Our explanation for this is that the PDX tumors pre-treated with paclitaxel are enriched for CSC or tumor initiating cells (similar to what we show in figure 5D where paclitaxel treatment increases the number of SP cells). Hence, the engraftment and outgrowth in the second-generation xenografts would be more efficient, resulting in larger tumors.

21. Line 217: Table S15 should be replaced with table S16.

The error has been corrected.

22. Line 218: Table S16 should be replaced with table S15.

The error has been corrected.

23. Line 223: “high FOXM1 expression in TNBC subtype”, it should be only “in TNBC subtype” according to table S18.

We would like to thank the reviewer for pointing out this mistake. The text has been changed accordingly.

24. Line 231: “Supplementary Figure S12C and S12D”. There are no such Figures in the supporting information. The authors are probably referring to Figure S7. The labeling in Figure S7 is incorrect as well.

The error has been corrected.

Reference:

1.         Emami, K.H., et al., A small molecule inhibitor of beta-catenin/CREB-binding protein transcription [corrected]. Proc Natl Acad Sci U S A, 2004. 101(34): p. 12682-7.

2.         Curtis, C., et al., The genomic and transcriptomic architecture of 2,000 breast tumours reveals novel subgroups. Nature, 2012. 486(7403): p. 346-52.

Reviewer 3 Report

The authors of this manuscript evaluated the role of CBP/B-catenin/FOXM1 pathway in triple-negative breast cancer (TNBC) and tested a CBP inhibitor ICG-001 as a potential therapeutic in preclinical models. Considering that TNBC is currently treated with chemotherapeutic agents, identification of novel therapeutic targets and discovery of less-toxic drugs is important.

Major concerns in the manuscript are listed below.

1.      The authors used different cell line for different experiments. Some experiments were conducted with MDA-MB-231, while others were conducted with MDA-MB-468 cells. Some were conducted with both. The authors did not explain the choice of different cell line in various experiments.

2.      All immunoprecipitation (IPs) studies were conducted 24 hours after treating the cells with ICG-001. Moreover, no input controls have been included. It is likely that ICG-001 inhibits the interaction of CBP and B-catenin by reducing the expression of either or both the proteins. Hence, instead of a reduction in interaction a loss of protein expression after 24 hours has been the contributing factor. Interactions can be observed as early as 1 hour. Hence, IP experiments need to be conducted for a short duration of treatment.

3.      Although ICG-001 was discovered in 2004, its clinical track-record could not be determined. Alternatively, a new specific CBP inhibitor (Cell Centric’s CCS1477) has been advanced to the clinic for the treatment of prostate cancer. It is unknown at this point whether ICG-001 failed to show any tumor growth inhibition in animal models due to poor pharmacokinetic (PK) properties or lack of efficacy. No CBP-target genes were measured in the tumor specimens to determine the reason for the lack of efficacy. The experiments need to be validated with CCS1477, if available.

4.      siRNA studies are conflicting. While FOXM1 and B-catenin siRNAs demonstrate strong inhibition of FOXM1 target gene expression, the effect of CBP siRNA was modest. One would expect that CBP, being downstream of B-catenin, should be the major player to affect the cellular properties.

5.      Control cell lines such as non-cancerous lines have not been used to demonstrate selectivity of ICG-001 on the proliferation of TNBC lines.

6.      The authors have not provided an explanation for the lack of efficacy of ICG-001 in vivo. Combination treatments were effective, while ICG-001 individual treatment failed to demonstrate any effect. It is unclear how ICG-001 worked in vitro, but not in vivo.

7.      The authors have also failed to provide any information on the patient-derived xenografts (PDXs) and PDX-derived cells.

8.      Overall, although the authors have attempted several models, the data did not drive home a clear conclusion on the importance of CBP in TNBC.

Author Response

Reviewer 3

We would like to thank the reviewer for evaluating our manuscript and provide constructive criticism and suggestions for improving the quality of the study. We are happy to address the concerns below.

1. The authors used different cell line for different experiments. Some experiments were conducted with MDA-MB-231, while others were conducted with MDA-MB-468 cells. Some were conducted with both. The authors did not explain the choice of different cell line in various experiments.

We used several triple TNBC cell lines in order to make the conclusions more generalizable. The in the case of CoIP experiments, we showed that treatment with ICG-001 reduces CBP/b-catenin binding in MDA-231, MBA-468 and SUM149. For CBP/FOMX1 binding we only validated the experiment in MDA-231 since we also had RNA-Seq data for this cell line. An effect of disrupting CBP/FOXM1-binding via ICG-001 was the reduction in FOXM1 protein levels, which we also show in MDA-468, albeit to a lesser degree.

2. All immunoprecipitation (IPs) studies were conducted 24 hours after treating the cells with ICG-001. Moreover, no input controls have been included. It is likely that ICG-001 inhibits the interaction of CBP and B-catenin by reducing the expression of either or both the proteins. Hence, instead of a reduction in interaction a loss of protein expression after 24 hours has been the contributing factor. Interactions can be observed as early as 1 hour. Hence, IP experiments need to be conducted for a short duration of treatment.

We agree with the observation made by the reviewer and have provided a 4h treatment time point showing the disruption of CBP/FOXM1 via ICG-001 in MDA-231 cells (new Figure 3A) after 4h and 24h. The data shows that protein levels of FOMX1 are not affected after a short treatment period, but the interaction with CBP is clearly reduced. We also have data showing that b-catenin levels are not affected. Although we didn’t look for changes in CBP protein levels, our lab has previously shown that treatment with ICG-001 does not affect CBP protein levels [1].

3. Although ICG-001 was discovered in 2004, its clinical track-record could not be determined. Alternatively, a new specific CBP inhibitor (Cell Centric’s CCS1477) has been advanced to the clinic for the treatment of prostate cancer. It is unknown at this point whether ICG-001 failed to show any tumor growth inhibition in animal models due to poor pharmacokinetic (PK) properties or lack of efficacy. No CBP-target genes were measured in the tumor specimens to determine the reason for the lack of efficacy. The experiments need to be validated with CCS1477, if available.

The clinical version of ICG-001 is PRI724, which has been is still in use in several clinical trials for Myeloid malignancies (NCT02828254), metastatic pancreatic carcinoma (NCT01764477), advanced solid tumors (NCT01302405). As we state below, ICG-001 is not considered to be a cytotoxic agent per se, but rather modulates the phenotype of cancer cells, specifically CSC-like cells as the current study as well as previous studies have shown (REF). There the viability studies have been conducted in combination with paclitaxel.

 4. siRNA studies are conflicting. While FOXM1 and b-catenin siRNAs demonstrate strong inhibition of FOXM1 target gene expression, the effect of CBP siRNA was modest. One would expect that CBP, being downstream of B-catenin, should be the major player to affect the cellular properties.

This is an interesting point raised by the reviewer. A possible explanation for the observed results could be that gene- or protein dosage of FOXM1 and b-catenin are more critical than CBP, but the interaction of CBP with FOXM1 and b-catenin is critical. Hence, while siRNA knockdown cannot fully eliminate CBP from the cells, enough of the protein might be left to drive CBP/FOMX1/b-catenin dependent gene expression. On the other hand, treatment with ICG-001 potently disrupts the binding of CBP to the transcriptional complex, thereby inhibiting gene expression. An explanatory section has been added to the discussion of our manuscript.

5. Control cell lines such as non-cancerous lines have not been used to demonstrate selectivity of ICG-001 on the proliferation of TNBC lines.

This is an important point by the reviewer. We would like to emphasize that ICG-001 is not considered a cell-killing agent. Rather, the proposed mechanism of action is that of a differentiating agent targeting cancer cells by modulating their phenotype, e.g. changing the CSC phenotype and/or rendering cancer cells less resistant to chemotherapy. Hence, the results in our study are primarily shown as a combination therapy of ICG-001 with paclitaxel. Nevertheless, the reviewer has pointed out an important issue regarding the effect on ICG-001 on cancer cell viability. Although not intended as a single agent, ICG-001 has some, albeit a moderate effect on CBP-dependent MDA-MB-231 cells, while not affecting non-cancerous MCF10a cells with low CBP-levels (Figure 1C). We included a supplementary figure showing the reduction in viable MDA-MB-231 cells but not MCF10a cells (Figure 1G). Regarding the moderate effect on TNBC cell viability we would like to stress again that ICG-001 is not intended to be used a single agent.

6. The authors have not provided an explanation for the lack of efficacy of ICG-001 in vivo. Combination treatments were effective, while ICG-001 individual treatment failed to demonstrate any effect. It is unclear how ICG-001 worked in vitro, but not in vivo.

ICG-001 is not a cytotoxic agent per se and does not kill tumor cells directly. Instead the proposed function is that of a modulating or differentiating agent, which leads to a change in CSC phenotype and related drug resistance. We demonstrate this by showing in vitro that ICG-001 strongly reduces the SP-phenotype and CSC-related gene expression (such as MDR1, ABCG2 and CD44). The observation that ICG-001 inhibits cell growth in vitro but not in vivo might be due the fact that cell cultures in vitro provide only a basic environment for cell growth, while in vivo an complex environment may provide additional growth cues for the cells. The important part of our study is that treatment with ICG-001 sensitizes previously resistant cells to paclitaxel and also inhibits the occurrence of CSC-like cells, which are thought to be responsible for tumor initiation and regrowth after initially successful treatment. We show this in vitro (Figure 5G) as well as in vivo by the secondary implantation of pre-treated tumors (Figure 7G).

7. The authors have also failed to provide any information on the patient-derived xenografts (PDXs) and PDX-derived cells.

Supplementary table S8 gives an overview of the origin of the human primary breast tumor tissues used for the PDX models.

8. Overall, although the authors have attempted several models, the data did not drive home a clear conclusion on the importance of CBP in TNBC.

We hope that the addition of new data requested by the reviewer as well as our explanations to the various led to a clearer picture and established the role of CBP in TNBC

Reference:

1.         Emami, K.H., et al., A small molecule inhibitor of beta-catenin/CREB-binding protein transcription [corrected]. Proc Natl Acad Sci U S A, 2004. 101(34): p. 12682-7.

Round  2

Reviewer 1 Report

This manuscript was revised, but my reply to the peer review is inadequate.

Author Response

We would like to thank the reviewer again for the time and effort to review our manuscript. Several changes have been made to latest version of our manuscript in order to improve upon the remaining issues.

Reviewer 2 Report

Unfortunately, this paper in the current state suffers from lack of a thorough  proofreading by the authors (same as the first submitted draft).

Some points to clarify:

1. Figure 1D: Which experiment does the graph showing the area under the curve refer to?  is it for MDA-MB-231, SUM-149 or MDA-MB-468 cells? Please also mention the type of probed protein (e.g., CBP or IgG)

2. Figure 3A: There is no blot for 24h treatment

3.  Figure 3C: Why are there two panels? What is the differences between these two panels?

4. Figure S3: the label and caption for Figure S3b should be removed since there is no such Figure.

5. Figure 5D: Y axis is missing a label

6. Lines 164-165: “Accordingly, tumors in mice bearing PDX1 showed more aggressive growth (Figure S4A) and worse response to therapy”.  It should be PDX2.

7. Figure 7B: Why does tumour volume significantly increase in the group treated with paclitaxel plus PBS as compared to one treated with PBS alone?

Authors reply: “Our explanation for this is that the PDX tumors pre-treated with paclitaxel are enriched for CSC or tumor initiating cells (similar to what we show in figure 5D where paclitaxel treatment increases the number of SP cells). Hence, the engraftment and outgrowth in the second-generation xenografts would be more efficient, resulting in larger tumors.”

If such explanation holds truth, why this phenomenon is not seen for Figures 7E and 7G (i.e., PDXs)?

Author Response

We would like to thank the reviewer again for taking the time and effort to read our manuscript and again provide useful critique. We are happy to address the concerns below point by point.

Unfortunately, this paper in the current state suffers from lack of a thorough proofreading by the authors (same as the first submitted draft).

The manuscript has been thoroughly proof read and mistakes have been corrected.

1. Figure 1D: Which experiment does the graph showing the area under the curve refer to?  is it for MDA-MB-231, SUM-149 or MDA-MB-468 cells? Please also mention the type of probed protein (e.g., CBP or IgG)

We would like to clarify that the aera under the curve refers to all three experiments, i.e. CoIP for all three cell lines. The summary results representing the reduction in CBP/b-catenin binding under DMSO (red bar) and ICG-001 (blue bar) treatment conditions are presented, showing a statistically significant reduction for all three cell lines (MDA-MB-231, MDA-MB-468 and SUM149). We have added a clarifying statement to the figure legend for Figure 1 (line 97 and 98) and highlighted the added section

2. Figure 3A: There is no blot for 24h treatment

We would like to thank the reviewer for pointing out that the plot is missing. The plot shown in the figure is for the 24h treatment (as was shown in the previous version of the figures). We have now added in a blot showing the 4h CoIP data for FOXM1 and CBP. The blot was cropped to fit the representation of the 24h treatment data.

3.  Figure 3C: Why are there two panels? What is the differences between these two panels?

We have edited and renumbered figure 3 to make it more clear what data is shown. The experiments were conducted using MDA-MB-231 and are as follows: Figure 3A represent CoIP data for the 4h treatment time point, Figure 3B shows the 24h treatment results. Figure 3D and 3E are western blots showing protein levels after 4h and 24h, respectively.

4. Figure S3: the label and caption for Figure S3b should be removed since there is no such Figure.

The mistake has been corrected.

5. Figure 5D: Y axis is missing a label

The mistake has been corrected. The label for the axis in this graph should read “Hoechst blue” for the X axis and “Hoechst red” for the Y axis.

6. Lines 164-165: “Accordingly, tumors in mice bearing PDX1 showed more aggressive growth (Figure S4A) and worse response to therapy”.  It should be PDX2.

The reviewer is correct and we apologize for the labeling error on our side. The mistake has been corrected.

7. Figure 7B: Why does tumour volume significantly increase in the group treated with paclitaxel plus PBS as compared to one treated with PBS alone?

Authors reply: “Our explanation for this is that the PDX tumors pre-treated with paclitaxel are enriched for CSC or tumor initiating cells (similar to what we show in figure 5D where paclitaxel treatment increases the number of SP cells). Hence, the engraftment and outgrowth in the second-generation xenografts would be more efficient, resulting in larger tumors.”

If such explanation holds truth, why this phenomenon is not seen for Figures 7E and 7G (i.e., PDXs)?

Again the reviewer has pointed out a quite interesting issue.

A statement has been added to the manuscript (line 296-306):

“While response to treatment in the primary tumors fit the proposed dependence on FOXM1 levels (i.e. PDX1 with low FOXM1 responded to paclitaxel alone while the MDA-MB-468 xenograft and PDX2 with higher levels of FOXM1 show the greatest reduction in tumor growth under the combination of paclitaxel with ICG-001), the behavior observed in the secondary outgrowth is more complex. Likely a more complex environment in vivo compared to in vitro culture affects the response to treatment. Treatment with ICG-001 resulted either in large tumors or prevented secondary outgrowth. It could be speculated that ICG-001 can either drive CSC towards a transiently amplifying phenotype that still has tumor initiating capacity but is more sensitive to paclitaxel treatment, or differentiate CSC to a level where these cells loose tumor initiating capacity. Especially in vivo this outcome might be determined by drug concentration within tissue, or modified by a complex tumor microenvironment.”

Additionally, we would like to add some further explanation in response to the above stated question:

PDX1 and PDX2 differ substantially in the expression of FOXM1. PDX1 shows a similar response to paclitaxel with or without the addition of ICG-001 in the treatment experiment (Figure 7D). Since the secondary outgrowth is reduced under both conditions and ICG-001 alone it could be speculated that in the FOXM1-low tumor barely any tumor initiating cells or CSC like cells exist (as shown in Figure 7C) and primary outgrowth is driven by a transiently amplifying cell population that is more sensitive to paclitaxel treatment or further differentiation by ICG-001. Hence, second generation outgrowth is significantly reduced (Figure 7E). Regarding PDX2, it is interesting to see that ICG-001 treatment resulted in the largest tumors. Again we propose that ICG-001 might result in the differentiation of CSC-like cells into transiently amplifying cells with much higher proliferation capacity, resulting in larger tumors. We believe that inherent differences exist in the growth of tumor cells in vitro and in vivo. Therefore, while ICG-001 inhibits the growth of cancer cells und simplified growth conditions in vitro (Figure 1G), in can enhance the outgrowth of tumors in a complex in vivo environment (Figure 7F), but could eventually exhaust this outgrowth once all CSC-like cells are differentiated. Future studies could look at serial transplantation of ICG-001 treated cells to further investigate this phenomenon. Treatment with paclitaxel seems to increase FOXM1 expression in vitro (Figure 6) and in vivo (Supplementary figure S4B) and CSC-like cells in vitro (Figures 5C and 5D), at this least in the cell line model that also shows the largest outgrowth in secondary xenografts (Figure 7B). The reason that PDX2 doesn’t behave like the cell line xenograft could be that CSC-like cells are not selected for to the same degree, perhaps do to the tumor microenvironment which is more complex in the PDX than in the cell line, composed of stromal cells and perhaps immune- or immunomodulatory cells. Perhaps treatment of the MDA-MB-468 xenograft results in more transiently amplifying cells and results in larger secondary outgrowth. These are all very interesting possibilities that could be explored in future studies

Reviewer 3 Report

I feel that the authors have satisfactorily addressed the comments. I don't have further comments. The manuscript can be accepted as is.

Author Response

We would like to thank the reviewer again for the time and effort to review our manuscript and to help to substantially improve it. Several changes have been made to the latest version to conform with requests made by other reviewers.